# An *in vitro* Study of Betadine’s Ability to Eliminate Live Bacteria on the Eye: Should It Be Used for Protection against Endophthalmitis?

**DOI:** 10.3390/antibiotics11111549

**Published:** 2022-11-04

**Authors:** Alyssa Nagle, Jonathan Kopel, John Reed, Unique Jacobo, Phat Tran, Kelly Mitchell, Ted W. Reid

**Affiliations:** Department of Ophthalmology and Visual Sciences, Texas Tech University Health Sciences Center, Lubbock, TX 79430, USA

**Keywords:** Betadine, antimicrobial, wounds, Gram-positive, Gram-negative

## Abstract

Background: Povidone-iodide (Betadine) is an antiseptic that is applied topically and has many uses in the medical community, such as in wound care and pre- and post-operative surgical procedures. This study was done to measure the effectiveness of Betadine solutions in inhibiting the growth of Gram-negative and Gram-positive bacteria. Methods: The ability of 2.5 and 10% Betadine solutions to inhibit bacterial growth was measured against *Staphylococcus aureus*, *Staphylococcus epidermidis*, *Pseudomonas aeruginosa*, and *Acinetobacter baumannii*. We grew the bacteria independently and together to simulate a hospital environment. Results: All the bacteria showed zones of inhibition. However, discs were also tested for live bacteria using the colony-forming unit assay. Complete killing was only seen for *S. aureus* with the 10% Betadine solution. All other bacteria showed growth on the disc. Conclusions: This study showed several things. First, the zone of inhibition assay does not give an accurate assessment of antimicrobial properties when used alone and should be followed by a colony-forming unit assay. Second, 2.5% and 5% Betadine do not have effective antimicrobial properties against any of the bacteria tested, and 10% Betadine is only effective against *S. aureus* and not effective against the other bacteria tested.

## 1. Introduction

This study was carried out to determine if Betadine is a good antimicrobial to use against endophthalmitis when the ocular barrier is penetrated or for surgical procedures [1,2,3,4]. The main bacterial species involved in the pathogenesis of endophthalmitis are *Staphylococcus* spp., and of that species, *S. epidermidis* was the highest isolate [5]. Betadine (povidone-iodine) is an antiseptic and antimicrobial that is used in both the treatment and prevention of bacterial growth in wounds and surgical sites [6,7,8,9]. Betadine is a complex that contains elemental iodine and polyvinylpyrrolidone (PVP-I). Since Betadine releases free iodine (a small molecule), it readily penetrates microbial membranes and oxidizes key materials within the cell, eventually leading to cell death [3]. Betadine is widely used to treat wounds because of its broad spectrum of antimicrobial properties against bacteria, fungi, amoeba, spores, protozoa, and several viruses [3,4]. Betadine is widely used for ophthalmologic procedures to decrease the rate of post-operative endophthalmitis [7,9,10,11]. Currently, concentrations of 5% Betadine solutions or higher are used as an antimicrobials in surgical procedures [12,13,14]. Concentrations below 5% are shown to be minimally effective. It is currently recommended to use at least 5%, if not higher. In 1989, the Centers for Disease Control reported contamination of Clonidine, a povidone-iodine solution for patients undergoing peritoneal dialysis, by the manufacturer [15]. A subsequent study in 2001 by Costertan and Stewart showed that the bacteria were capable of growing in a PVP-I bottle [16]. Yet, PVP-I is commonly used in ophthalmology for a variety of procedures. We have recently shown that betadine is ineffective in killing some bacteria [17]. Yet, due to its continued use in ophthalmology, we sought to determine the efficacy of PVP-I in preventing the growth of bacteria found in endophthalmitis. Thus, we carried out experiments to determine the ability of PVP-I to kill bacteria and inhibit growth on a surface.

## 2. Materials and Methods:

### 2.1. Povidone-Iodide Solution Preparation

We used 10% povidone-iodine (PVP-I) or Betadine topical bactericide (#NDC 67618-150-05, distributed by Avrio Health LP. Stanford, CT, USA 06901-3431). Betadine solution was diluted to 2.5% and 5% in sterile 1X PBS and stored at room temperature.

### 2.2. Bacterial Strains, Media, and Growth Conditions

The laboratory bacterial strains tested were *S. aureus* GFP (green fluorescent protein) strain AH133 and *P. aeruginosa* PAO1 GFP strain, both of which express a green fluorescent protein from plasmids pCM11 and pMRP9-1, respectively [18,19,20]. Plasmids expressing green fluorescent protein (GFP) were introduced into *S. aureus* AH133 (pCM11) and PAO1 (pMRP9-1) to allow the visualization of biofilms in vitro. We used the staphylococcus and pseudomonas strains because these bacteria are the most commonly found bacteria for endophthalmitis. Erythromycin at 10 μg/mL and carbenicillin at 300 μg/mL were used to maintain the plasmids in AH133 and PAO1, respectively [18,19,20]. The strains were grown in Luria broth (LB) at 37 °C with shaking (150 rpm) using the Thermo Scientific Thermolyne SHKE4000 Incubator Shaker. The LB media was created in our lab using sodium chloride (# S4444, Teknova, Hollister, CA, USA), trypton (#BP1421-2, Fisher Scientific, Waltham, MA, USA) and yeast extract (#Y9050, Teknova, Hollister, CA, USA). In order to maintain pCM11 in AH133, 1 μg/mL of erythromycin was added to the LB. To maintain pMRP9-1 in PAO1, 300 μg/mL of carbenicillin was added to the LB. The clinical isolates used in this study were *S. epidermidis* and *A. baumannii*. The efficacy of the Betadine solutions was examined using LB broth medium and LB agar, pseudomonas isolation agar, and aureus ChromoSelect agar as the growth mediums. The clinical isolates were obtained from the clinical lab at Texas Tech University Health Sciences Center under an approved Institutional Review Board protocol (IRB # L13-034, 2020), Texas Tech University Medical Center/Lubbock, TX, USA.

### 2.3. Zone of Inhibition (ZOI) Assay

Bacteria were grown overnight in LB broth medium at 37 °C with shaking (150 rpm). The next day, the bacterial suspensions were adjusted to an OD_600_ of 0.5 in PBS. After this, a sterile cotton swab was dipped into the adjusted bacterial culture, and a lawn of bacteria was made on LB agar plates using the swab. The Betadine discs were prepared by adding 20 μL of 2.5 and 10% Betadine solution onto 6 mm diameter BBL blank paper discs. Three Betadine discs were used for each concentration. The three discs were distributed evenly onto the LB agar. The plates were then incubated at 37 °C for 24 h to allow for growth. The results were then read and recorded. The diameter of the zones was measured to the nearest millimeter, including the diameter of the BBL discs.

### 2.4. Colony Forming Unit (CFU) Assay on the Betadine Treated Disc

Following the zone of inhibition assay, the remaining live bacteria on the discs were quantified using the CFU assay. Following the 24-h incubation, each disc was moved from the agar plates to a sterile 1.5 mL microcentrifuge tube containing 1 mL of PBS. The tubes were vortexed 3 times for 1 min to detach the cells from the discs. The now suspended cells were serially diluted in PBS to a 6-fold dilution, and 10 μL aliquots of each dilution were spotted onto agar plates.

### 2.5. Confocal Laser Scanning Microscopy (CLSM)

The discs for the CLSM were prepared as described above in the Material and Methods for the zone of inhibition assay. Three untreated control and three Betadine discs for both concentrations of Betadine were examined for live bacteria that remained on the discs. CLSM of the *S. aureus* GFP AH133 and *P. aeruginosa* GFP PAO1 was done using a Nikon Eclipse Ni-E upright confocal laser scanning microscope (Nikon, Melville, NY, USA). The images were processed and analyzed using NIS-Elements Imaging Software. The lab strains of *S. aureus* GFP AH133/pCM11 and *P. aeruginosa* GFP PAO1/pMRP9-1 express green fluorescent protein from plasmids pCM11 and pMRP9-1 when grown with 1 µg/mL erythromycin or 300 µg/mL carbenicillin, respectively.

### 2.6. Bacterial Combination Assay

Bacteria were grown overnight in a LB medium at 37 °C with shaking (150 rpm). The bacterial suspensions were individually adjusted to an OD_600_ of 0.5 in a phosphate-buffered saline (PBS) solution. Following this, 1 mL of each bacterial culture was combined in a sterile tube. A lawn was prepared, and the zone of inhibition assay was then conducted, as outlined above. Following the zone of inhibition assay, a CFU assay was also conducted as outlined above. The bacteria were isolated by plating CFUs on LB agar, pseudomonas isolation agar, and aureus ChromoSelect agar.

### 2.7. Zero-Hour Biofilm Assay

*S. aureus* was grown overnight in an LB medium at 37 °C with shaking (150 rpm). The bacterial suspension was adjusted to an OD_600_ of 0.5 in a phosphate-buffered saline (PBS) solution. Following this, bacterial lawns were made on 2 LB agar plates, and 3 discs containing PBS for the control and 3 discs containing 5% Betadine for the treatment group were placed on separate plates. The plates were then incubated for 24 h at 37 °C. The next day, a CFU assay was executed, as outlined above.

### 2.8. 24-Hour Biofilm Assay

*S. aureus* was grown overnight in an LB medium at 37 °C with shaking (150 rpm). The bacterial suspension was adjusted to an OD_600_ of 0.5 in a phosphate-buffered saline (PBS) solution. Following this, bacterial lawns were made on 2 LB agar plates, and 3 blank discs were placed on each of the plates. The plates were then incubated for 24 h at 37 °C. The following day, the blank discs were removed from the plates and placed into two separate sterile Petri dishes. PBS was added to one set for control, and 5% Betadine was added to the other set for treatment. The two sets were then added to separate blank LB agar plates and incubated for 24 h at 37 °C. The following day, a CFU assay was conducted, as outlined above.

## 3. Results

As seen in Figure 1A and Figure 2A, all bacteria, when grown individually and together, showed ZOIs of greater than 6 mm (diameter of the cellulose disc) when tested at both Betadine concentrations. The diameter of the disc is represented by the line drawn in Figure 1A and Figure 2A. The results of the CFU assay (Figure 1B and Figure 2B) showed different results than those obtained from the ZOI assay. The 2.5% Betadine solution did not completely kill any of the bacteria that were tested, with *P. aeruginosa* and *A. baumannii* showing no difference in the viable count compared to the control trial. *S. epidermidis* even showed more growth than the control trial when grown with *S. aureus*.

The 10% Betadine solution only showed the complete killing of *S. aureus* when grown individually and with *S. epidermidis*. *S. epidermidis* showed a lower viable count at 10% when grown with *S. aureus*, but still showed incomplete killing. *P. aeruginosa* showed no difference from the control trial viable count when grown individually and only a slightly lower count when grown with *A. baumannii*; however, incomplete killing still occurred. *A. baumannii* showed a lower viable count when grown individually and with *P. aeruginosa* but did not show complete killing.

The confocal laser scanning microscopy results confirmed what was found with the CFU results for *S. aureus* GFP AH133 and *P. aeruginosa* PAO1 GFP (these are the only bacteria that carried the GFP gene for detection in these experiments). As seen in Figure 3, only the 10% Betadine solution completely killed *S. aureus* GFP AH133, and incomplete killing at 2.5% was seen when grown individually and with *S. epidermidis* CI. As seen in Figure 4, 10% Betadine only partially killed *P. aeruginosa* PAO1 GFP when grown individually and with *A. baumannii CI*.

Since *Staphylococcus aureus* was the only bacteria that was completely killed by the 10% Betadine solution, a zero- and 24-h biofilm assay was done to test the ability of a 5% Betadine solution to kill *S. aureus*. This was conducted because 5% Betadine is the recommended concentration to be used during surgical procedures. As seen in Figure 5, 5% Betadine was not able to completely kill *S. aureus* in either biofilm assay. For the zero-hour biofilm assay, there was a small decrease in the viable count from the control to the discs treated with 5% Betadine. There was little difference in the viable counts for the control and treated discs for the 24-h biofilm.

## 4. Discussion

Betadine is a widely used antiseptic and antimicrobial in many ophthalmology procedures and other surgical procedures by applying it to the surface of the eye, a skin wound, or as a prep for surgery [1,2,3,4,21,22,23]. The discomfort in patients after applying betadine in the eye is infrequent; however, some patients report a burning sensation [24,25,26]. Some investigators that report a low tolerability in patients suggest lowering the concentration of the Betadine solution [24,27]. Recently, articles have suggested that PVP-I is a potent broad-spectrum antimicrobial. In contact, as seen in Figure 1 and Figure 2, it is not effective as either an antiseptic or an antimicrobial [1,2,3]. Thus, even though the incidents of infection are infrequent, the use of Betadine needs to be re-examined.

One result, as seen in Figure 1 and Figure 2, is that the zone of inhibition assay does not accurately represent the ability of an antibacterial to kill bacteria. This is based on our observation that even though we observed zones of inhibition, the discs still produced viable colonies in our CFU assay. This is best seen with the 10% Betadine solution, which showed zones of inhibition for all bacteria. However, it only showed the complete killing of *S. aureus*, while *P. aeruginosa* and *A. baumannii* showed no or little change in the viable count from the control trial and the 2.5% Betadine solution. This is an important observation because the ZOI assay has been used in many studies to determine the ability of antibacterials to kill different strains of bacteria.

In addition, we found that even 10% Betadine is not sufficient to kill most of the bacteria that we studied (with the exception of *S. aureus*). We also found that when *S. aureus* was treated with 5% Betadine, the recommended concentration for surface preparation, it was not sufficient to kill even this bacteria. We can therefore conclude that 5% Betadine would not be sufficient to kill any of the other bacteria tested since the higher concentration of 10% was not sufficient to kill them. In addition, since the 5% concentration is not able to kill a zero time lawn of *S. aureus*, this lack of killing is not due to biofilm formation. All of the bacteria used in this study are strains that can be found growing in eyes infected with endophthalmitis. Therefore, these data indicate that Betadine is an insufficient antimicrobial for use in ophthalmology procedures. Other treatments should be considered.

## 5. Conclusions

This study showed several things. First, the zone of inhibition assay does not give an accurate assessment of antimicrobial properties when used alone and should be followed by a colony-forming unit assay. Second, 2.5% and 5% Betadine do not have effective antimicrobial properties against any of the bacteria tested. Lastly, 10% Betadine is only effective against *S. aureus* and not effective against the other bacteria tested. Thus, it would appear that Betadine is a poor choice for removing bacteria from the eye or on the skin before an eye injection or operation procedure.

## Figures and Tables

**Figure 1 antibiotics-11-01549-f001:**
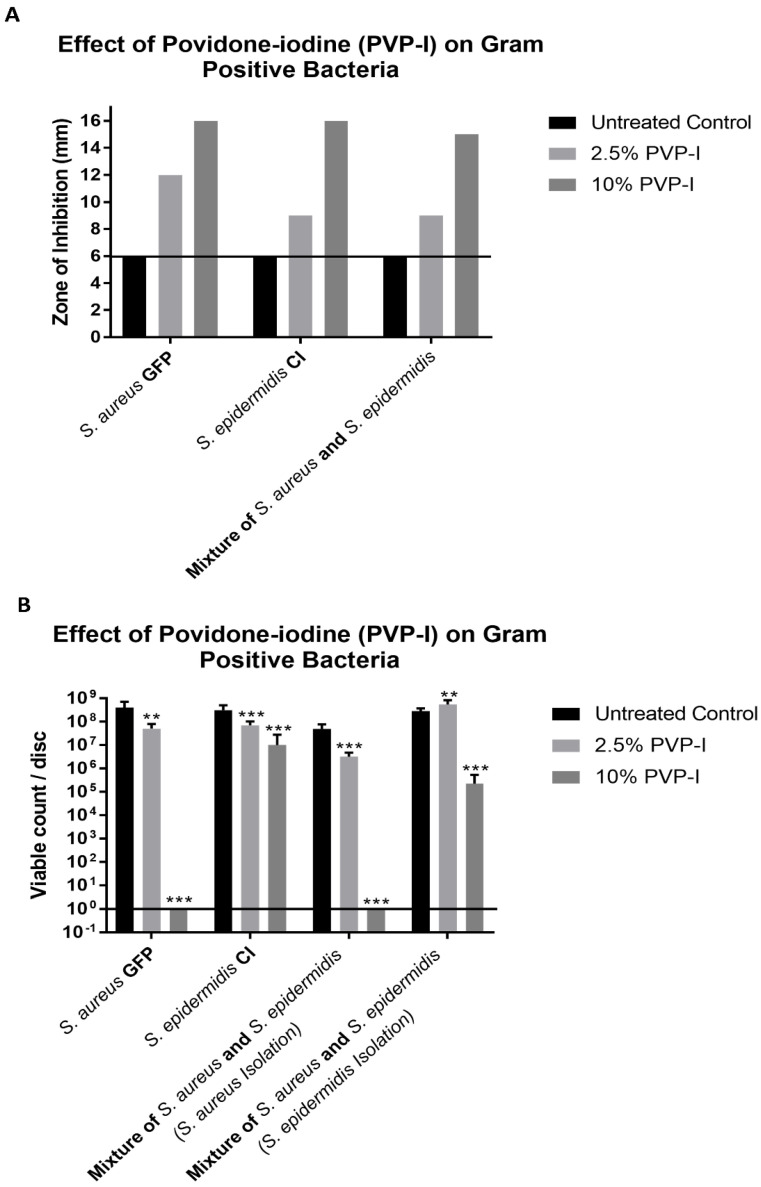
(**A**) The zones of inhibition (ZOI) of Gram-positive bacteria on the discs were measured in mm. (**B**) The bacteria remaining on the disc was quantified by the CFU assay. The different numbers of stars represent significance: ** *p* < 0.01, and *** *p* < 0.001.

**Figure 2 antibiotics-11-01549-f002:**
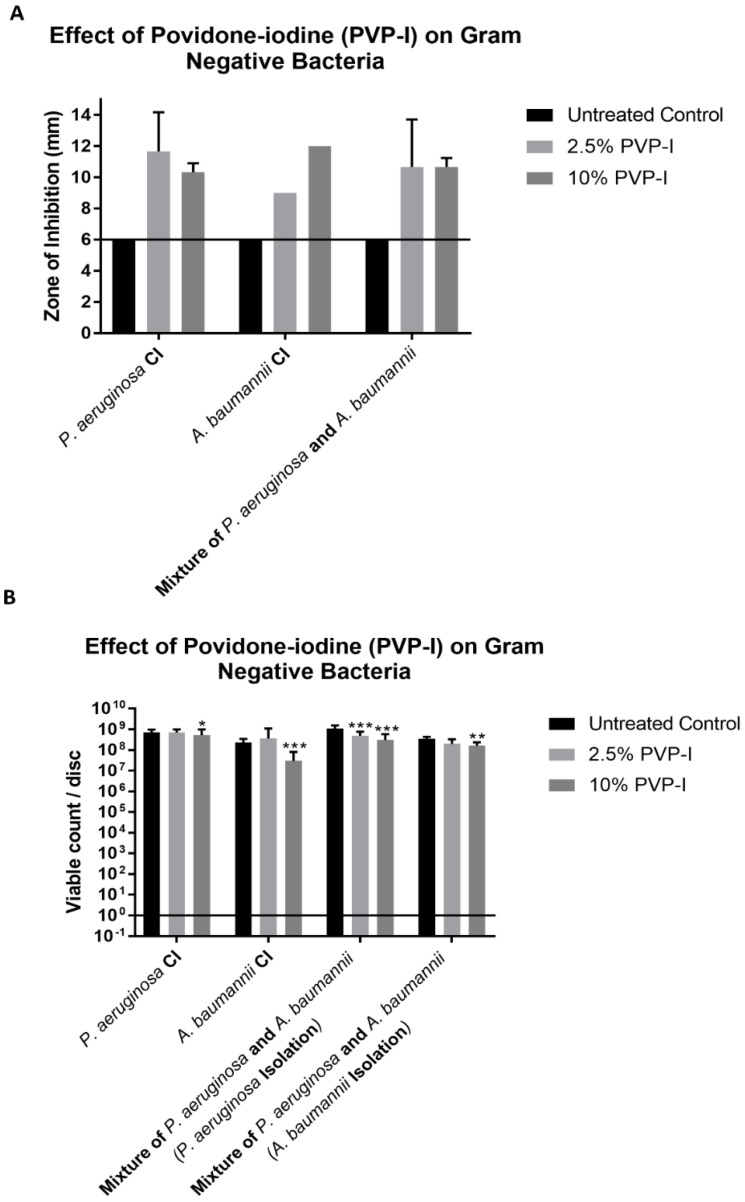
(**A**) The ZOI of gram-negative bacteria on the discs were measured in mm. (**B**) The bacteria remaining on the disc was quantified by the CFU assay. The different numbers of stars represent significance: * *p* < 0.05, ** *p* < 0.01, and *** *p* < 0.001.

**Figure 3 antibiotics-11-01549-f003:**
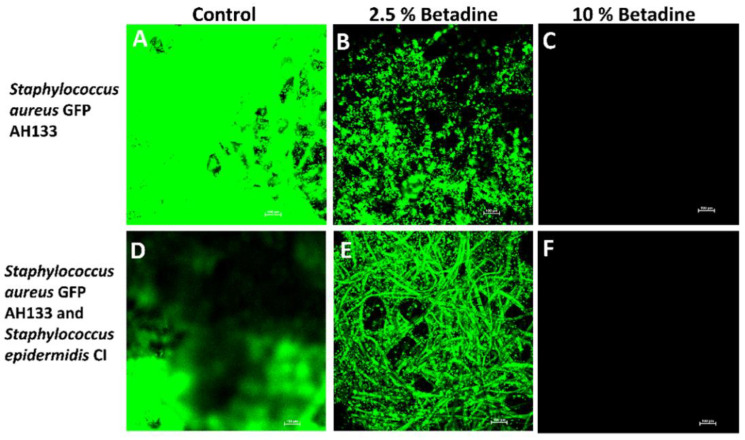
Confocal laser scanning microscopy of the *S. aureus* GFP AH133 grown alone (**A**–**C**) and *S. aureus* GFP AH133 and *S. epidermidis* CI grown together (**D**–**F**) that remained on the disc after treatment.

**Figure 4 antibiotics-11-01549-f004:**
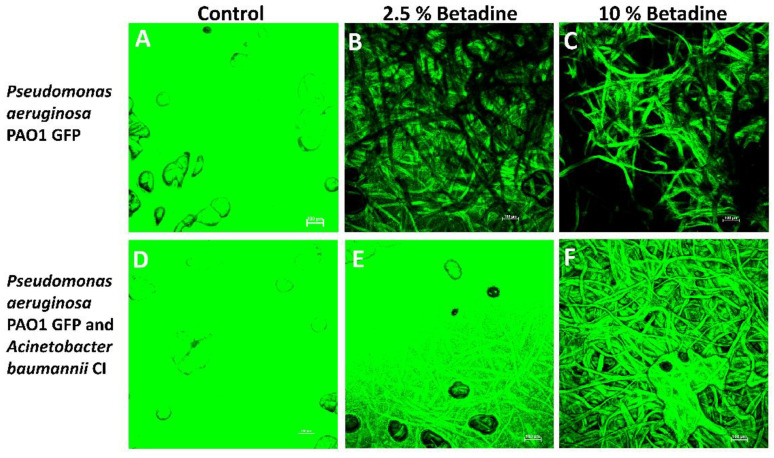
Confocal laser scanning microscopy of the *P. aeruginosa* PAO1 GFP grown alone (**A**–**C**) and *P. aeruginosa* PAO1 GFP and *A. baumannii* CI grown together (**D**–**F**) that remained on the disc after treatment with Betadine.

**Figure 5 antibiotics-11-01549-f005:**
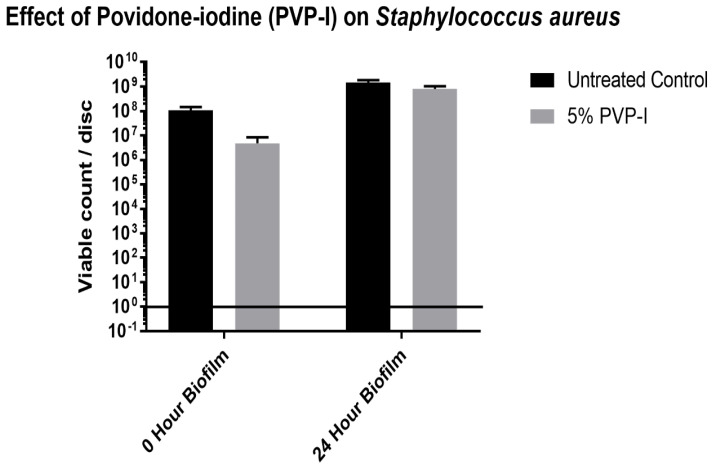
*S. aureus* that remained on discs quantified by CFU assay after treatment with 5% PVP-I, for both a zero time and a 24 h biofilm.

## Data Availability

Not applicable.

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
