# Peer review of "An in vitro Study of Betadine’s Ability to Eliminate Live Bacteria on the Eye: Should It Be Used for Protection against Endophthalmitis?"

_antibiotics, 2022, doi:10.3390/antibiotics11111549_

Round 1

Reviewer 1 Report

From the title, the manuscript "An in vitro Study of Betadine's Ability to Inhibit the Growth of Gram Negative and Gram Positive Bacteria: Should We Use it on a Wound?" promises a very interesting study because Betadine 10% is a pharmaceutical product currently widely used in medical units and in-home care, as such or diluted. The authors performed valuable advanced antibacterial studies: first - on media with a single bacteria species and second - with 2 associations of two bacteria growing together (Gram-positive and Gram-negative, separately), proving the limits of Betadine's antibacterial activity.

However, I have some comments for the authors. 

A. Major 

Introduction:

The "Introduction" is too short and does not contain the aim of the present study. The authors described Betadine, its antibacterial mechanism, showed its use in ophthalmology, and indicated the recommended concentrations. Here, the introduction is finished. 

They should present background about the bacteria implied in ophthalmic infections and a short history regarding their susceptibility or resistance to Betadine. Maybe the bacteria selected are known as responsible for these infections. Thus, the reader will be introduced step by step in their study. They should indicate if other authors previously performed such observations, or maybe they are the first. Trying to understand their notations about tested bacterial strains, I found two studies published as abstracts in 2014 and 2015 by the same last author with other co-authors:

https://iovs.arvojournals.org/article.aspx?articleid=2332660

articleid=2271421https://iovs.arvojournals.org/article.aspx?

One of these co-authors observed and reported these phenomena in 2014-2015. They could specify it. The present study is more extensive and appears to be based on previously mentioned ones.

In the final of "Introduction," the authors must present the aim of their present study. They can shortly explain their notations that follow the scientific name of bacteria species. Here, or in materials and method, in the correspondent section, with suitable references. A small number of readers are familiarized with bacterial fluorescence and plasmids. 

Material and methods:

The authors did not provide any references regarding the techniques used. For example, DDM was adapted from CLSI, and the following methods were performed according to other previously published studies. They are not very common, and more data about them are necessary. They should also indicate the provenance of their study's culture media and other chemicals (Section 4.2.). On the other hand, in the same section, the authors mentioned that the clinical isolates used in this study were Staphylococcus epidermidis and Acenitobacter baumannii (it is better Acinetobacter, according to scientific literature), and obtained from the Clinical lab at Texas Tech University Health Sciences Center under an approved Institutional Review Board protocol, Texas Tech University Medical center/Lubbock, Texas, 132 USA. They should indicate the provenance of Staphylococcus aureus GFP strain AH133 122 and Pseudomonas aeruginosa PAO1 GFP strain MM294, which express a green fluorescent protein from plasmids pCM11 and pMRP9-1, respectively. 

The authors stated: "Bacteria are grown overnight in a LB medium at 37⁰C with shaking (250 rpm - line 125, and 150 rpm - lines 135, 162, 170, 177)" but did not mention what LB medium is and the apparatus that allows such conditions for bacterial growth (incubation and shaking).

Results:

For better understanding, the explanations of Figures 1 and 2 should have the letters A and B placed at the beginning or final of the entire explication (not in the middle of the proposition, where A was placed). Moreover, all signs in both figures should be explained (there were marked in the attached manuscript).

Regarding Figures 3 and 4 with CLSM images, I think more details are needed for a straightforward interpretation: only S. aureus and P. aeruginosa have fluorescence, or have all 4 bacteria tested?

Discussion: 

In "Discussion," the authors comment on their results, mixing discussion and conclusions (lines 97 and 109).

The succession of used techniques is motivated by their aim to clearly show that the zone of inhibition assay (Disc Diffusion Method - DDM) does not give a realistic account of the ability of an antibacterial to kill bacteria.

Their study appears so interesting; in my opinion, it is better to justify why they opted for the tested bacteria species.

The title appears to be general, not focused only on ophthalmology; if they want to remain in this form, they must generalize their study application, proving, with suitable references, the implications of bacteria tested in other various pathologies. 

They should discuss, in detail, the interesting succession of their experiments, step by step. In addition, they must compare all obtained results with other studies from the scientific literature, especially the previously published ones regarding Betadine's antibacterial activity on Acinetobacter sp, P. aeruginosa, and S. epidermidis.

For example, I suggest some interesting studies:

Ruben Barreto, Brigitte Barrois, Julien Lambert, Surbhi Malhotra-Kumar, Victor Santos-Fernandes, Stan Monstrey, Addressing the challenges in antisepsis: focus on povidone iodine, International Journal of Antimicrobial Agents, Volume 56, Issue 3, 2020; https://doi.org/10.1016/j.ijantimicag.2020.106064.

Castelnuovo S: Povidone Iodine 0.66% to Fight Pseudomonas aeruginosa in Contact Lens Wearer: A Case Report. Case Rep Ophthalmol 2022:398-407. doi: 10.1159/000524539

Parvin, F.; Vickery, K.; Deva, A.K.; Hu, H. Efficacy of Surgical/Wound Washes against Bacteria: Effect of Different In Vitro Models. Materials 2022, 15, 3630. https://doi.org/10.3390/ ma15103630 

Houang ET, Gilmore OJ, Reid C, Shaw EJ. Absence of bacterial resistance to povidone iodine. J Clin Pathol. 1976 Aug;29(8):752-5. doi: 10.1136/jcp.29.8.752. PMID: 821972; PMCID: PMC476160.

A case report about a Contaminated Povidone-Iodine Solution - Texas is available online at https://www.cdc.gov/mmwr/preview/mmwrhtml/00001358.htm

Uygur, F., Özyurt, M., Evinç, R. et al. Comparison of octenidine dihydrochloride (Octenisept®), polihexanide (Prontosan®) and povidon iodine (Betadine®) for topical antibacterial effects in Pseudomonas aeruginosa-contaminated, full-skin thickness burn wounds in rats. cent.eur.j.med 3, 417–421 (2008). https://doi.org/10.2478/s11536-008-0042-x

and a recently published article with the same co-author as an innovative synergic solution for wounds:

Phat Tran, Jonathan Kopel, Keaton Luth, Huy Dong, Ameesh Dev, Dilip Mehta, Kelly Mitchell, Keith W. Moeller, Cameron D. Moeller, Ted Reid, The in vitro efficacy of betadine antiseptic solution and colloidal silver gel combination in inhibiting the growth of bacterial biofilms, American Journal of Infection Control, 2022, https://doi.org/10.1016/j.ajic.2022.04.002. 

Thus, they can highlight the novelty of their research. 

Conclusions:

After developing "Discussions," supported and compared with suitable references, the authors could resume the essential points about the antibacterial effects of betadine against the bacteria tested in a few phrases. 

B. Minor comments and suggestions can be found in the attached manuscript.

Author Response

Reviewer #1

From the title, the manuscript "An in vitro Study of Betadine's Ability to Inhibit the Growth of Gram Negative and Gram Positive Bacteria: Should We Use it on a Wound?" promises a very interesting study because Betadine 10% is a pharmaceutical product currently widely used in medical units and in-home care, as such or diluted. The authors performed valuable advanced antibacterial studies: first - on media with a single bacteria species and second - with 2 associations of two bacteria growing together (Gram-positive and Gram-negative, separately), proving the limits of Betadine's antibacterial activity.

However, I have some comments for the authors. 

Introduction:

The "Introduction" is too short and does not contain the aim of the present study. The authors described Betadine, its antibacterial mechanism, showed its use in ophthalmology, and indicated the recommended concentrations. Here, the introduction is finished.  Additions were made to the introduction including the aims of the study.

They should present background about the bacteria implied in ophthalmic infections and a short history regarding their susceptibility or resistance to Betadine. Maybe the bacteria selected are known as responsible for these infections. Thus, the reader will be introduced step by step in their study. They should indicate if other authors previously performed such observations, or maybe they are the first. Trying to understand their notations about tested bacterial strains, I found two studies published as abstracts in 2014 and 2015 by the same last author with other co-authors:

https://iovs.arvojournals.org/article.aspx?articleid=2332660

articleid=2271421https://iovs.arvojournals.org/article.aspx?

These are not articles and are abstracts for a meeting.  Thus, the enclosed article is more complete and contains the data proving the point that betadine is a poor choice as an antiseptic and antimicrobial.

One of these co-authors observed and reported these phenomena in 2014-2015. They could specify it. The present study is more extensive and appears to be based on previously mentioned ones.

In the final of "Introduction," the authors must present the aim of their present study. They can shortly explain their notations that follow the scientific name of bacteria species. Here, or in materials and method, in the correspondent section, with suitable references. A small number of readers are familiarized with bacterial fluorescence and plasmids. 

We thank the reviewer for their comments. We added the following sentence to answer their critique on the focus of the study: “In 2001, it was shown by Costertan and Stewart that bacteria was capable of growing in PVP-I bottle. Yet, PVP-I is commonly used in ophthalmology for a variety of procedures. Yet, due to its continued use in ophthalmology, we sought to determine the efficacy of PVP-I in preventing the growth of bacteria in endo-ophthalmitis”. We added a paper on the type of bacteria involved in endo-ophthalmitis. We could not find any papers discussing the resistance to PVP-I.  We also provided background on the endophthalmitis.

Material and methods:

The authors did not provide any references regarding the techniques used. For example, DDM was adapted from CLSI, and the following methods were performed according to other previously published studies. They are not very common, and more data about them are necessary. They should also indicate the provenance of their study's culture media and other chemicals (Section 4.2.). On the other hand, in the same section, the authors mentioned that the clinical isolates used in this study were Staphylococcus epidermidis and Acenitobacter baumannii (it is better Acinetobacter, according to scientific literature), and obtained from the Clinical lab at Texas Tech University Health Sciences Center under an approved Institutional Review Board protocol, Texas Tech University Medical center/Lubbock, Texas, 132 USA. They should indicate the provenance of Staphylococcus aureus GFP strain AH133 122 and Pseudomonas aeruginosa PAO1 GFP strain MM294, which express a green fluorescent protein from plasmids pCM11 and pMRP9-1, respectively. 

The authors stated: "Bacteria are grown overnight in a LB medium at 37⁰C with shaking (250 rpm - line 125, and 150 rpm - lines 135, 162, 170, 177)" but did not mention what LB medium is and the apparatus that allows such conditions for bacterial growth (incubation and shaking).

We appreciate the reviewer’s comment. We provided references for the techniques used. We also indicated where we received the cells used through the references provided. We also provided the information on the shaker used in the experiment

Results:

For better understanding, the explanations of Figures 1 and 2 should have the letters A and B placed at the beginning or final of the entire explication (not in the middle of the proposition, where A was placed). Moreover, all signs in both figures should be explained (there were marked in the attached manuscript).

Regarding Figures 3 and 4 with CLSM images, I think more details are needed for a straightforward interpretation: only S. aureus and P. aeruginosa have fluorescence, or have all 4 bacteria tested?

We thank the reviewer for their comments. We made the appropriate changes to the Figures 1 and 2. In addition, we clarified in the paper that we only looked at the S. aureus and P. aeruginosa since there were the only bacteria that contained the green fluorescent protein.

Discussion: 

In "Discussion," the authors comment on their results, mixing discussion and conclusions (lines 97 and 109).

We took out the word conclusions and compared results from what we previously did. We also added two sentences to help with this transition and comparison.

The succession of used techniques is motivated by their aim to clearly show that the zone of inhibition assay (Disc Diffusion Method - DDM) does not give a realistic account of the ability of an antibacterial to kill bacteria. Their study appears so interesting; in my opinion, it is better to justify why they opted for the tested bacteria species.

We made sure to mention that we tested the S. aureus and P. pseudomonas species since they are the most common bacteria found in endo-ophthalmitis.  Also, S. epidermidis is the most common isolate and is the most resistant to betadine.

The title appears to be general, not focused only on ophthalmology; if they want to remain in this form, they must generalize their study application, proving, with suitable references, the implications of bacteria tested in other various pathologies. 

We changed the title focus on eye wounds primarily to focus the paper. We added six references to answer this comment, which include the ones given by the reviewer.

They should discuss, in detail, the interesting succession of their experiments, step by step. In addition, they must compare all obtained results with other studies from the scientific literature, especially the previously published ones regarding Betadine's antibacterial activity on Acinetobacter sp, P. aeruginosa, and S. epidermidis.

Using the previous comments, we made use to compare and discuss our results as well as include previous studies compared to our own.

For example, I suggest some interesting studies:

Ruben Barreto, Brigitte Barrois, Julien Lambert, Surbhi Malhotra-Kumar, Victor Santos-Fernandes, Stan Monstrey, Addressing the challenges in antisepsis: focus on povidone iodine, International Journal of Antimicrobial Agents, Volume 56, Issue 3, 2020; https://doi.org/10.1016/j.ijantimicag.2020.106064.

Castelnuovo S: Povidone Iodine 0.66% to Fight Pseudomonas aeruginosa in Contact Lens Wearer: A Case Report. Case Rep Ophthalmol 2022:398-407. doi: 10.1159/000524539

Parvin, F.; Vickery, K.; Deva, A.K.; Hu, H. Efficacy of Surgical/Wound Washes against Bacteria: Effect of Different In Vitro Models. Materials 2022, 15, 3630. https://doi.org/10.3390/ ma15103630 

Houang ET, Gilmore OJ, Reid C, Shaw EJ. Absence of bacterial resistance to povidone iodine. J Clin Pathol. 1976 Aug;29(8):752-5. doi: 10.1136/jcp.29.8.752. PMID: 821972; PMCID: PMC476160.

A case report about a Contaminated Povidone-Iodine Solution - Texas is available online at https://www.cdc.gov/mmwr/preview/mmwrhtml/00001358.htm

Uygur, F., Özyurt, M., Evinç, R. et al. Comparison of octenidine dihydrochloride (Octenisept®), polihexanide (Prontosan®) and povidon iodine (Betadine®) for topical antibacterial effects in Pseudomonas aeruginosa-contaminated, full-skin thickness burn wounds in rats. cent.eur.j.med 3, 417–421 (2008). https://doi.org/10.2478/s11536-008-0042-x

and a recently published article with the same co-author as an innovative synergic solution for wounds:

Phat Tran, Jonathan Kopel, Keaton Luth, Huy Dong, Ameesh Dev, Dilip Mehta, Kelly Mitchell, Keith W. Moeller, Cameron D. Moeller, Ted Reid, The in vitro efficacy of betadine antiseptic solution and colloidal silver gel combination in inhibiting the growth of bacterial biofilms, American Journal of Infection Control, 2022, https://doi.org/10.1016/j.ajic.2022.04.002. 

 Conclusions:

After developing "Discussions," supported and compared with suitable references, the authors could resume the essential points about the antibacterial effects of betadine against the bacteria tested in a few phrases. 

We thank the reviewer for their comment. We took their advice and made the appropriate changes to the conclusions to address this critique.

Reviewer 2 Report

This work described in detail the killing effect of betadine on various clinical strains of bacteria that are a significant problem in ophthalmology. The research was carried out carefully and clearly documented. The paper describes only the results of microbiological tests carried out in vitro, and I have not doubts for this part.  However, I have some comments on the general conclusion of the article. Authors did not refer to the overall effect of betadine on the whole organism. It should be remembered that there are many immunomodulatory mechanisms, the anti-inflammatory effect, free radical and antioxidant reactions, which means that in a living organism these processes take place differently and the effectiveness of the preparation is therefore different from that described in vitro. Please read for example International Journal of Surgery 44 (2017) 260e268 http://dx.doi.org/10.1016/j.ijsu.2017.06.073.

I consider the final conclusions premature because they only concern in vitro tests and only a single aspect of betadine action.

All the above limitations should be mentioned in the work. Moreover, in the discussion, please refer to the previously obtained results and their interpretation. Also the title should be changed as the current one does not reflect the content of the work.

Minor comment: lines 92-93; Please change the sentence.

After the above changes I’ll recommend the work for publication.

Author Response

Reviewer #2

This work described in detail the killing effect of betadine on various clinical strains of bacteria that are a significant problem in ophthalmology. The research was carried out carefully and clearly documented. The paper describes only the results of microbiological tests carried out in vitro, and I have not doubts for this part.  

However, I have some comments on the general conclusion of the article. Authors did not refer to the overall effect of betadine on the whole organism. It should be remembered that there are many immunomodulatory mechanisms, the anti-inflammatory effect, free radical and antioxidant reactions, which means that in a living organism these processes take place differently and the effectiveness of the preparation is therefore different from that described in vitro. Please read for example International Journal of Surgery 44 (2017) 260e268 http://dx.doi.org/10.1016/j.ijsu.2017.06.073.

We thank the author for their comment. We want to clarify that we are only looking at the effect of betadine on bacteria living on the surface of the eye and not the bacteria that translocate to the interior of the eye.

I consider the final conclusions premature because they only concern in vitro tests and only a single aspect of betadine action.

We thank the review. We want to emphasize that this study was to examine if betadine was effective against bacteria on the surface, not internally.

All the above limitations should be mentioned in the work. Moreover, in the discussion, please refer to the previously obtained results and their interpretation. Also the title should be changed as the current one does not reflect the content of the work.

We thank reviewer for the comments. We made the appropriate changes to the discussion as per reviewer 1. In addition, we modified the title as suggested by reviewer 1.

Minor comment: lines 92-93; Please change the sentence.

We made the changes to the manuscript.

Reviewer 3 Report

The manuscript by Alyssa Nagle et al. describes the impact of Betadine in inhibiting the growth of Gram-negative and Gram-positive bacteria.

The manuscript suffers from methodological shortcomings, the reading is not clear, and it seems necessary to rework the entire work before hoping for a possible publication.

The title needs to be changed as the work does not evaluate wound involvement.

Given the scope of the manuscript, the addition of a microbiologist author seems necessary.

It is not appropriate to do such a methodology to evaluate the impact of an antiseptic, the authors should consider setting up bactericidal curves.

Prefer passive turns.

Betadine is not an antibiotic but an antiseptic.

The selection criteria for clinical strains used should be more specific.

The number of the IRB opinion must be specified.

Has the reading of the inhibition diameters been done by several different authors? The inter-operator variability exists and is major in view of the differences observed.

Paragraphs 4.5 and 4.6 need to be extensively reviewed to be understood.

It is not very understandable to study resistance profiles by growing the strains on selective media. Please justify.

Authors contributions : Final approval of the manuscript must be shared by all authors and not by the last one only.

Author Response

Reviewer #3

The manuscript by Alyssa Nagle et al. describes the impact of Betadine in inhibiting the growth of Gram-negative and Gram-positive bacteria.The manuscript suffers from methodological shortcomings, the reading is not clear, and it seems necessary to rework the entire work before hoping for a possible publication.

We thank the reviewer for their observation. We made changes to the methodology, readability, and clarifications to the previous aforementioned areas of the paper as suggested by the previous reviewers.

The title needs to be changed as the work does not evaluate wound involvement.

We made the appropriate changes to the review as suggested by the previous reviewers

Given the scope of the manuscript, the addition of a microbiologist author seems necessary.

We thank the reviewer for their suggestions. Dr. Tran’s doctoral degree is in microbiology.  Thus, he is a microbiologist.

It is not appropriate to do such a methodology to evaluate the impact of an antiseptic, the authors should consider setting up bactericidal curves.

The study evaluates the ability of betadine to act as both an antiseptic and antimicrobial. Since the main purpose of betadine’s use is to kill bacteria so no living bacteria make it into the eye, it is its antimicrobial ability that matters most. However, we show that betadine is a poor antiseptic and antimicrobial.

Prefer passive turns.

We thank the reviewer for this comment. The appropriate changes were made to the manuscript.

Betadine is not an antibiotic but an antiseptic.

We thank the reviewer for their comment. However, betadine is used as an antimicrobial on the eye.

The selection criteria for clinical strains used should be more specific.

These strains are the most common bacteria found in endo-ophthalmitis and appropriate references were added to the manuscript.

The number of the IRB opinion must be specified.

We thank the reviewer for their comment. The study did not use any human specimens or patients. Therefore, no IRB opinion was warranted.

Has the reading of the inhibition diameters been done by several different authors? The inter-operator variability exists and is major in view of the differences observed.

We thank the reviewer for this comment. Only one person did the interpretation for the inhibition diameters.

Paragraphs 4.5 and 4.6 need to be extensively reviewed to be understood.

We thank the reviewer for this comment. We made the appropriate changes to the language, readability, and clarifications as well as references in these paragraphs.

It is not very understandable to study resistance profiles by growing the strains on selective media. Please justify.

We thank the reviewer for this comment. The only selective compound added to the media was so that the GFP gene would remain within the bacteria.

Authors contributions : Final approval of the manuscript must be shared by all authors and not by the last one only.

We thank the reviewer for this comment. We made the appropriate changes to the paper since all authors shared in final approval.

Round 2

Reviewer 1 Report

Thank you for your efforts to rectify your manuscript according to my comments from Round 1.

Several aspects appeared in the revised manuscript or remained unsolved; they are displayed in the following comments: 

1. Line 43: the correct form is: "endophthalmitis are Staphylococcus spp. and of that species, S. epidermidis" - please check and correct.

2. Line 62: "preventing the growth of bacteria in found in endophthalmitis"-  please delete the first "in."

3. Line 71-72: AH133 and PAO1 are internal notations for S. aureus and P. aeruginosa with GFP? Please, explain in the text their significance.

4. Lines 83-84: Please, mention the provider for culture media (name, city, state, and country). If they are internal media, please mention that.

5. Essential documents are still missing:

5.1. Line 85-87 - The authors stated: "The clinical isolates were obtained from the Clinical lab at 85 Texas Tech University Health Sciences Center under an approved Institutional Review 86 Board protocol, Texas Tech University Medical center/Lubbock, Texas, USA." (Please, add the registration number and date).

5.2. Lines 229-230: "All of the bacteria used in this study are strains that can 229 be found growing in eyes infected with enthopthalmitis." (minor comment: Please, correct the last word).

5.3. On the other hand, in lines 251-252, they wrote: "Institutional Review Board Statement: Not applicable." The same observation for "Informed Consent Statement: Not applicable."

5.4. Please, provide both relevant documents with registering number and date.

6. Figures 1 and 2:  lines 175 and 185-186: Please, remove "and" after (B). In lines 176 and 187: Please correct as follows: the statistical significance:  * p<0.05, **p<0.01, and ***p<0.001 and remove "such that."

7. Lines 192 and 196: It is better (D, E, and F) without space between E and F. Please correct. Considering the uniformity, you can write similarly: (A, B, and C).

8. Lines 229 and 231: Please, remove the spaces between successive phrases. 

As an overview, the manuscript is not edited following the MDPI Instructions for Authors - the sections and subsections are un-numbered, and their titles are not written suitably; I have the same observation regarding the References. The entire text of the manuscript must be rigorously checked for bacterial species names still written without italics (as in the abstract) in the keywords: Gram-positive and Gram-negative (please, correct); for the text uniformity: if the name is entirely written once (for example, Staphylococcus aureus), in the rest of the text the abbreviation should be currently used (S. aureus). 

Author Response

Reviewer #1:

Thank you for your efforts to rectify your manuscript according to my comments from Round 1.

Several aspects appeared in the revised manuscript or remained unsolved; they are displayed in the following comments: 

  1. Line 43: the correct form is: "endophthalmitis are Staphylococcusspp. and of that species, S. epidermidis" - please check and correct.

We thank the reviewer for their comment. We made the change to the grammar as suggested.

  1. Line 62: "preventing the growth of bacteria in found in endophthalmitis"- please delete the first "in."

We thank the reviewer for their comment. We made the change to the grammar as suggested.

  1. Line 71-72: AH133 and PAO1 are internal notations for S. aureusand P. aeruginosa with GFP? Please, explain in the text their significance.

We appreciate the reviewer’s comment. AH133 represents a staph aureus strain in our study. The PAO1 stands for Pseudomonas Aeruginosa Strain 1. The GFP stands for green fluorescent protein, which was used for visualization to compliment our colony forming assays. We used the staphylococcus ssp. and pseudomonas strains because these bacteria are the most commonly found bacteria for endophthalmitis. We put references below for clarity. We did not try to do an exhaustive study. Many references are there if desired.

Predictors of Post-Injection Endophthalmitis: A Multivariable Analysis Based on Injection Protocol and Povidone Iodine Strength

Maxwell S. Stem, Prethy Rao, Ivan J. Lee, Maria A. Woodward, Lisa J. Faia, Jeremy D. Wolfe, Antonio Capone, Jr., Douglas Covert, A. Bawa Dass, Kimberly A. Drenser, Bruce R. Garretson, Tarek S. Hassan, Alan Margherio, Kean T. Oh, Paul V. Raephaelian, Sandeep Randhawa, Scott Sneed, Michael T. Trese, Sunita Yedavally, George A. Williams, Alan J. Ruby

Ophthalmol Retina. 2019 May; 3(5): 456.  Ophthalmol Retina. 2019 Jan; 3(1): 3–7.

Durand ML (2017). “Bacterial and Fungal Endophthalmitis.” Clin Microbiol Rev 30(3): 597–613. [PMC free article] [PubMed] [Google Scholar] [Ref list]

The majority of North American EE cases are a result of infection with S. aureus 

Jackson TL, Paraskevopoulos T and Georgalas I (2014). “Systematic review of 342 cases of endogenous bacterial endophthalmitis.” Surv Ophthalmol 59(6): 627–635. [PubMed] [Google Scholar].

The main gram positive found was S. aureus.  One of the main gram negative bacteria was Psedudomonas aeruginosa

Clin Microbiol Rev. 2017 Jul; 30(3): 597–613.

Published online 2017 Mar 29. doi:  PMCID: PMC5475221 PMID: 28356323

Bacterial and Fungal Endophthalmitis

Marlene L. Durand*

The microbiology of postinjection endophthalmitis includes coagulase-negative staphylococci (65%), 

Journal of Ophthalmology Volume 2016 | Article ID 6764192 | https://doi.org/10.1155/2016/6764192

Show citation

Causative Microorganisms of Infectious Endophthalmitis: A 5-Year Retrospective Study

Fang Duan,1Kaili Wu,1Jingyu Liao,1Yongxin Zheng,1Zhaohui Yuan,1Junlian Tan,1and Xiaofeng Lin

Specifically, the main causative organisms for infections were all Staphylococcus spp. (31.4%, 33 of 105); among them, Staphylococcus epidermidis (15.2%, 16 of 105) was the predominant isolate.

  1. Lines 83-84: Please, mention the provider for culture media (name, city, state, and country). If they are internal media, please mention that.

We made the LB broth from scratch in our lab. We used sodium chloride (# S4444, Teknova, Hollister, CA), trypton (#BP1421-2, Fisher scientific, Waltham, MA) and Yeast extract (#Y9050, Teknova, Hollister, CA)

  1. Essential documents are still missing:

5.1. Line 85-87 - The authors stated: "The clinical isolates were obtained from the Clinical lab at 85 Texas Tech University Health Sciences Center under an approved Institutional Review 86 Board protocol, Texas Tech University Medical center/Lubbock, Texas, USA." (Please, add the registration number and date).

We appreciate the reviewer’s comment. The IRB umber and date was given

5.2. Lines 229-230: "All of the bacteria used in this study are strains that can 229 be found growing in eyes infected with enthopthalmitis." (minor comment: Please, correct the last word).

We thank the reviewer for their comment. We made the change to the grammar as suggested.

5.3. On the other hand, in lines 251-252, they wrote: "Institutional Review Board Statement: Not applicable." The same observation for "Informed Consent Statement: Not applicable."

We appreciate the reviewer’s comment. The IRB umber and date was given for these statements. The statements were also updated as well.

5.4. Please, provide both relevant documents with registering number and date.

These are clinical isolates from patient samples. These are not ATCC cell lines. Lab strains tend to lose their virulency over time. Patient samples are more robust and model real world bacterial strains found in the clinic.

  1. Figures 1 and 2:  lines 175 and 185-186: Please, remove "and" after (B). In lines 176 and 187: Please correct as follows: the statistical significance:  *p<0.05, **p<0.01, and ***p<0.001 and remove "such that."

We thank the reviewer for their comment. The appropriate changes were made to the manuscript.

  1. Lines 192 and 196: It is better (D, E, and F) without space between E and F. Please correct. Considering the uniformity, you can write similarly: (A, B, and C).

We thank the reviewer for their comment. The appropriate changes were made to the manuscript.

  1. Lines 229 and 231: Please, remove the spaces between successive phrases. 

We thank the reviewer for their comment. The appropriate changes were made to the manuscript.

As an overview, the manuscript is not edited following the MDPI Instructions for Authors - the sections and subsections are un-numbered, and their titles are not written suitably; I have the same observation regarding the References.

We made sure to number the subsections of the manuscript and change the reference style to the appropriate MDPI style in endnote.

 The entire text of the manuscript must be rigorously checked for bacterial species names still written without italics (as in the abstract) in the keywords: Gram-positive and Gram-negative (please, correct); for the text uniformity: if the name is entirely written once (for example, Staphylococcus aureus), in the rest of the text the abbreviation should be currently used (S. aureus). 

We thank the reviewer for their comments. We went ahead and double checked the language and abbreviations used through the text, especially with regards to bacteria.

Reviewer 2 Report

I accept the manuscript in present form

Author Response

We thank the reviewer for their comment

Reviewer 3 Report

Some modifications have been considered for the revised version of the manuscript.

Nevertheless, major remains, and justify the need for rejection of the manuscript.

It is not appropriate to do such a methodology to evaluate the impact of an antiseptic, the authors should consider setting up bactericidal curves.

The study evaluates the ability of betadine to act as both an antiseptic and antimicrobial. Since the main purpose of betadine’s use is to kill bacteria so no living bacteria make it into the eye, it is its antimicrobial ability that matters most. However, we show that betadine is a poor antiseptic and antimicrobial.

--> This comment was not adequately addressed and the limitation remains.

Betadine is not an antibiotic but an antiseptic.

We thank the reviewer for their comment. However, betadine is used as an antimicrobial on the eye.

--> The use of betadine outside its usual field of use does not make it an antibiotic. Limit yourself to calling it an antiseptic (which has a definition).

The selection criteria for clinical strains used should be more specific.

These strains are the most common bacteria found in endo-ophthalmitis and appropriate references were added to the manuscript.

--> The selection of the strain did not provide sufficient justification for the selection of these particular strains, and therefore the selection criteria should be detailed.

Has the reading of the inhibition diameters been done by several different authors? The inter-operator variability exists and is major in view of the differences observed.

We thank the reviewer for this comment. Only one person did the interpretation for the inhibition diameters.

--> This could be considered as a major limitation.

Author Response

Reviewer #3:

Some modifications have been considered for the revised version of the manuscript.

Nevertheless, major remains, and justify the need for rejection of the manuscript.

It is not appropriate to do such a methodology to evaluate the impact of an antiseptic, the authors should consider setting up bactericidal curves.

The study evaluates the ability of betadine to act as both an antiseptic and antimicrobial. Since the main purpose of betadine’s use is to kill bacteria so no living bacteria make it into the eye, it is its antimicrobial ability that matters most. However, we show that betadine is a poor antiseptic and antimicrobial.

--> This comment was not adequately addressed and the limitation remains.

We did not do bactericidal curves because we only looked at the most common concentrations of betadine used by ophthalmologists.

Betadine is not an antibiotic but an antiseptic.

We thank the reviewer for their comment. However, betadine is used as an antimicrobial on the eye.

--> The use of betadine outside its usual field of use does not make it an antibiotic. Limit yourself to calling it an antiseptic (which has a definition).

We have looked at the literature. Some scientists characterize betadine as an antibiotic and some as an antiseptic. The definition of the function of betadine is not universal. References are shown below that show that betadine is an antimicrobial. However, our data shows that betadine is also a poor antimicrobial and antiseptic agent. Our assays did examine the antiseptic and antimicrobial activity of betadine. We measured the growth of the bacteria onto the disc, which would measure the antiseptic ability of betadine.

However, as we stated earlier, from an ophthalmologic standpoint, the ophthalmologist is not concerned whether betadine is an antimicrobial or an antiseptic. As long as there are no live bacteria that enters the eye, that is the overall objective of the ophthalmologist using betadine on the eye before surgery or injection into the eye. This is the reason why ophthalmologists use betadine on the eye. Thus, we wanted to focus on betadine’s ability to eliminate live bacteria overall. 

Antimicrob Agents Chemother. 2020 Sep; 64(9): e00682-20.

Published online 2020 Aug 20. Prepublished online 2020 Jun 22. doi: 10.1128/AAC.00682-20

PMCID: PMC7449185

PMID: 32571829

Povidone Iodine: Properties, Mechanisms of Action, and Role in Infection Control and Staphylococcus aureus Decolonization

Didier Lepelletier,a Jean Yves Maillard,b Bruno Pozzetto,c,d and Anne Simone

PVP-I is a water-soluble iodophor (or iodine-releasing agent) that consists of a complex between iodine and a solubilizing polymer carrier, polyvinylpyrrolidone (Fig. 1) (23, 24). In aqueous solution, a dynamic equilibrium occurs between free iodine (I2), the active bactericidal agent, and the PVP-I-complex. 

In vivo antimicrobial activity of 0.6% povidone-iodine eye drops in patients undergoing intravitreal injections: a prospective study

Daniele Tognetto,#1 Marco R. Pastore,#1 Lorenzo Belfanti,1 Riccardo Merli,1 Alex L. Vinciguerra,1 Marina Busetti,2 Giulia Barbati,3 and Gabriella Cirigliano1

Treatment with 0.6% povidone-iodine eye drops significantly reduced the conjunctival bacterial load from baseline (p < 0.001 for blood agar and p < 0.001 for chocolate agar) with an eradication rate of 80%. 

Comparison of the antimicrobial efficacy of povidone-iodine-alcohol versus chlorhexidine-alcohol for surgical skin preparation on the aerobic and anaerobic skin flora of the shoulder region

Dorothea Dörfel, Matthias Maiwald, Georg Daeschlein, Gerald Müller, Robert Hudek, Ojan Assadian, Günter Kampf, Thomas Kohlmann, Julian Camill Harnoss, Axel Kramer

Antimicrob Resist Infect Control. 2021; 10: 17. Published online 2021 Jan 22. doi: 10.1186/s13756-020-00874-8

PMCID: PMC7821636

The selection criteria for clinical strains used should be more specific.

These strains are the most common bacteria found in endo-ophthalmitis and appropriate references were added to the manuscript.

--> The selection of the strain did not provide sufficient justification for the selection of these particular strains, and therefore the selection criteria should be detailed.

We thank the reviewer for the comment. We looked into the literature and found several references that show that staph aureus, staph epidermidis, and pseudomonas are the most common bacterial strains found in endopthalmitis. However, we included other strains that are also found in the references below. We did not try to do an exhaustive study. Many references are there if desired. We only wanted to show that betadine does not work on the most common bacteria associated with endopthalmitis. Those minor bacteria will be examined in future studies.

Predictors of Post-Injection Endophthalmitis: A Multivariable Analysis Based on Injection Protocol and Povidone Iodine Strength

Maxwell S. Stem, Prethy Rao, Ivan J. Lee, Maria A. Woodward, Lisa J. Faia, Jeremy D. Wolfe, Antonio Capone, Jr., Douglas Covert, A. Bawa Dass, Kimberly A. Drenser, Bruce R. Garretson, Tarek S. Hassan, Alan Margherio, Kean T. Oh, Paul V. Raephaelian, Sandeep Randhawa, Scott Sneed, Michael T. Trese, Sunita Yedavally, George A. Williams, Alan J. Ruby

Ophthalmol Retina. 2019 May; 3(5): 456.  Ophthalmol Retina. 2019 Jan; 3(1): 3–7.

Durand ML (2017). “Bacterial and Fungal Endophthalmitis.” Clin Microbiol Rev 30(3): 597–613. [PMC free article] [PubMed] [Google Scholar] [Ref list]

The majority of North American EE cases are a result of infection with S. aureus 

Jackson TL, Paraskevopoulos T and Georgalas I (2014). “Systematic review of 342 cases of endogenous bacterial endophthalmitis.” Surv Ophthalmol 59(6): 627–635. [PubMed] [Google Scholar].

The main gram positive found was S. aureus.  One of the main gram negative bacteria was Psedudomonas aeruginosa

Clin Microbiol Rev. 2017 Jul; 30(3): 597–613.

Published online 2017 Mar 29. doi:  PMCID: PMC5475221 PMID: 28356323

Bacterial and Fungal Endophthalmitis

Marlene L. Durand*

The microbiology of postinjection endophthalmitis includes coagulase-negative staphylococci (65%), 

Journal of Ophthalmology Volume 2016 | Article ID 6764192 | https://doi.org/10.1155/2016/6764192

Show citation

Causative Microorganisms of Infectious Endophthalmitis: A 5-Year Retrospective Study

Fang Duan,1Kaili Wu,1Jingyu Liao,1Yongxin Zheng,1Zhaohui Yuan,1Junlian Tan,1and Xiaofeng Lin

Specifically, the main causative organisms for infections were all Staphylococcus spp. (31.4%, 33 of 105); among them, Staphylococcus epidermidis (15.2%, 16 of 105) was the predominant isolate.

Has the reading of the inhibition diameters been done by several different authors? The inter-operator variability exists and is major in view of the differences observed.

We thank the reviewer for this comment. Only one person did the interpretation for the inhibition diameters.

--> This could be considered as a major limitation.

We double checked and found that we had two people check the inhibition diameters for these experiments (Alicia and Phat)